METHODS

# CASPULE: A computational tool to study sticker spacer polymer condensates

**Aniruddha Chattaraj**, **David S. Kanovich**, **Srivastav Ranganathan**,
**Eugene I. Shakhnovich** *

Department of Chemistry and Chemical Biology, Harvard University, Cambridge, Massachusetts, United States of America

* shakhnovich@chemistry.harvard.edu

## Abstract

Phase separated condensates are recognized as a ubiquitous mechanism of spatial organization in cell biology. Biophysical modeling of condensates provides critical insights into the dynamics and functions of these subcellular structures that are difficult to extract via experiments. Here we present an efficient computational pipeline, CASPULE (**C**ondensate **A**nalysis of **S**ticker Spacer **P**olymers **U**sing the **L**AMMPS **E**ngine), to simulate and analyze the biological condensates made of sticker-spacer polymers. CASPULE implements a unique force field that combines traditional Langevin dynamics with a "detailed-balance proof" protocol for single-valent bond formation between stickers. This framework allows us to study the non-trivial biophysics that emerge from single-valent sticker interactions, coupled with the effect of separating the energetic contributions of stickers and spacers. We provide detailed documentation on how to setup the simulation environment, perform simulations and analyze the results. Through case studies, we highlight the utility and efficacy of our pipeline. Importantly, we provide statistical parameters to characterize the cluster size distribution often observed in biological systems. We envision this tool to be broadly useful in decoding the interplay of kinetics and thermodynamics underlying the formation and function of biological condensates.

## Author summary

Living cells contain droplet-like compartments called biomolecular condensates that organize important molecules and reactions without a surrounding membrane. Understanding how these condensates form and behave helps us learn more about healthy cellular function and diseases. Condensate forming biopolymers (proteins and nucleic acids) often contain a sticker spacer architecture that gives rise to interesting emergent properties when they assemble into a condensate. Here we report an easy-to-use computational pipeline, called CASPULE, that allows us to simulate and analyze the condensation of polymer

**Data availability statement:** An extensive documentation and related source codes are available at https://caspule.github.io/caspule/.

**Funding:** This work was supported by the National Institutes of Health Grant 5R35GM139571 to EIS. The funders had no role in study design, data collection and analysis, decision to publish, or preparation of the manuscript.

**Competing interests:** The authors have declared that no competing interests exist.

chains. Unlike previous tools, CASPULE models both the physical motion of these molecules and specific bonding interactions between sticker segments. This user-friendly tool will enable researchers to explore the governing principles underlying the formation and function of biomolecular condensates.

## Introduction

Biomolecular condensates are membraneless, phase-separated compartments that organize and regulate diverse biochemical processes across the cytoplasm, nucleus, and cellular membranes [1–4]. Condensates are now widely recognized as viscoelastic network fluids that often arise when phase separation is coupled to percolation (PSCP) in multivalent, sticker–spacer macromolecules [5]. In this view, stickers form transient, *specific* physical crosslinks while spacers tune solubility, chain flexibility, and the coupling between density and networking transitions. This modular architecture endows condensates with rheological behaviors and kinetic responses that differ sharply from those of simple homopolymer fluids, motivating computational frameworks that can connect molecular parameters to mesoscale organization and dynamics.

In many sticker–spacer systems, each sticker can bind only one partner at a time (strict 1:1 valency). This single-occupancy constraint changes condensate kinetics, topology, and rheology in ways that are not reproduced by purely isotropic, short-range attractions (e.g., Lennard Jones forces). For example, a polymer model with stickers that form at most one specific bond can result in an exhaustion of free valencies within nascent clusters. This in turn produces metastable, long-lived multi-droplet states even without active processes [6]. This kinetic arrest is not captured by simple isotropic attractions because it depends on the book-keeping of available bonding sites and on valency-limited crosslinking within and between clusters [7].

Patchy-particle simulations [8] likewise demonstrate that valency and patch topology reorganize condensates across scales, from multilayered interiors to differential client/scaffold partitioning. In a minimal model, Sanchez-Burgos et al. [9] show that higher-valency species preferentially populate condensate cores while lower-valency species enrich interfaces. Additionally, Polyansky et al. [10] connect interaction valency and protein compactness to the fractal dimension of condensates, providing an analytical bridge from atomistic contacts to droplet-scale structure. These trends, which reflect network-connectivity maximization under valency constraints, further underscore that single-site occupancy (not just attraction strength) is a first-order control knob. Furthermore, studies also suggest that the nature of interactions matters for regulation of condensates. Using in-vitro experiments and patchy-particle simulations, Ghosh et al. [11] establish three archetypal classes of macromolecular regulators -- volume-exclusion promoters, weak-attraction suppressors, and strong-attraction promoters. The classification is an outcome of site-resolved interactions which depend on the relative strengths of specific versus non-specific

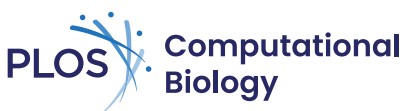

attractions which reiterate the need to model specific and non-specific contributions separately rather than by a single iso-tropic interaction term. Taken together, these studies show that systems with strict 1:1 sticker binding can phase-separate yet display distinct cluster statistics, slowed coarsening, metastable multi-droplet states, multilayered composition, and different viscoelastic responses compared with systems modeled by simple isotropic attractions. Therefore, the incorpora-tion of single-valent and anisotropic specific interactions is critical in simulations aimed at predicting dynamic properties, assembly kinetics and architectures.

Current simulation tools cover parts of this landscape but leave a practical gap. Lattice-based Monte Carlo engines such as LASSI [12] can map equilibrium phase behavior for multivalent polymers efficiently. However, they do not furnish time-resolved dynamics of cluster growth, aging, and coalescence. On the other hand, despite capturing the dynamics, conventional coarse-grained MD, such as OpenABC [13] or Martini-3 [14], typically lacks a principled, efficient mechanism to enforce single-valent sticker binding. The existing software, such as Readdy [15] or SpringSaLaD [16], serves as stand-alone applications to model polymer crosslinking; but they are limited in terms of the system size due to the lack of parallel execution.

Here we introduce CASPULE (**C**ondensate **A**nalysis of **S**ticker Spacer **P**olymers **U**sing the **L**AMMPS **E**ngine), a pipeline that enforces strict 1:1 site valency while preserving physical polymer dynamics. CASPULE can leverage the broad and diverse capabilities of LAMMPS [17], a coarse-grained MD engine extremely popular amongst multi-ple scientific communities. Within LAMMPS, CASPULE encodes chain connectivity via harmonic bonds and bending elasticity, non-specific cohesion via truncated Lennard-Jones interactions with tunable well depth (Ens), and specific sticker-sticker bonds via a shifted-harmonic well of depth Es that forms and breaks stochastically. Critically, each sticker can hold at most one bond at a time, ensuring single-valent, reversible crosslinking. This design decouples the energet-ics of specific binding (Es) from weaker non-specific (Ens) interactions, enabling cleaner titrations of distinct interaction types.

We formalize CASPULE as an end-to-end, reproducible protocol and provide reference tests that (a) highlight how Es and Ens independently tune network connectivity and density transitions, (b) validate that dynamics obeys detailed balance, and thus converges to canonical distribution, (c) map phase boundaries in parameter space using standardized readouts (energy traces, sticker-bond saturation, cluster statistics), and (d) display scalability of our approach.

## Materials and methods

The overall workflow of CASPULE contains three steps – setup, simulate and analyze (Fig 1). We have provided stepwise instructions in our CASPULE website (https://caspule.github.io/caspule/). For convenience, we will summarize the key steps here.

### A.  Step 1 – Simulation setup

**A.1 Construction of sticker spacer polymer:** First, we define a building block consisting of stickers and spacers. This block segment is then repeatedly combined into a chain of arbitrary length where the number of segments define the length of chain, and arrangement of sticker-spacer within a segment determines the overall spatial pattern of the chain. This method allows us to create chains with arbitrary lengths and patterns.

**A.2 Setting up data file:** Once multiple chain types are created, we use Moltemplate [18] and PACKMOL [19] to create the initial condition of the simulation. Moltemplate allows us to instantiate multiple copies of a given chain, and PACKMOL provides initial coordinates of those chain beads that can be fit into a simulation box prescribed by the user. The initial condition is stored as a "data" file.

**A.3 Force-field:** We use the LAMMPS package [17] to perform Langevin Dynamics simulations of our sticker spacer poly-mers. An input file contains all the force-field information to model polymer movement and structure.

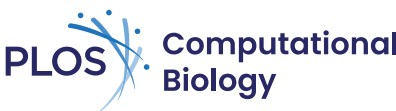

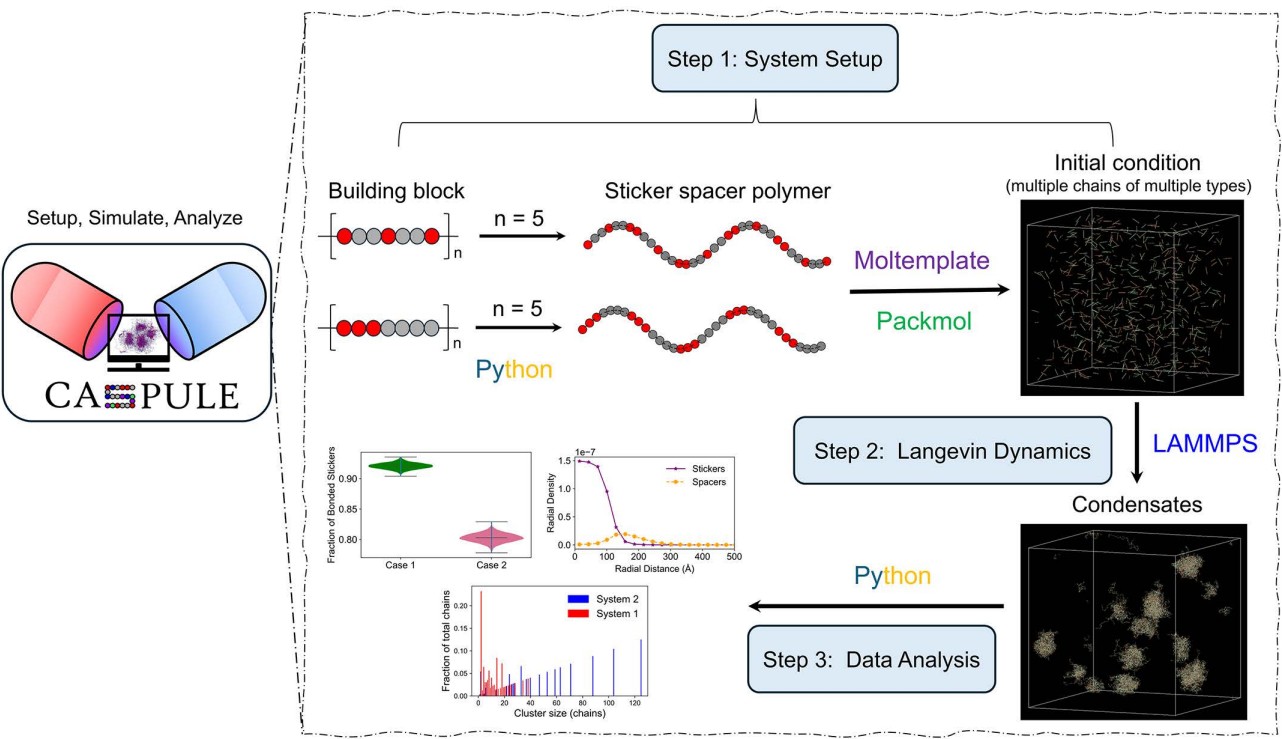

**Fig 1. Overview of the CASPULE pipeline.** It has three components – setup (step 1), simulation (step 2) and analysis (step 3). In step 1, we create a sticker spacer polymer chain in a template-based manner, where a pattern of sticker-spacer serves as a building block. Multiple repeats of such blocks create a chain. Red and grey beads represent stickers and spacers, respectively. Once the chains are created, we pack multiple copies inside the simulation box using the Moltemplate and PACKMOL software packages. This creates an initial condition which may contain many copies of multiple chain types. In step 2, we simulate the polymer condensation using the LAMMPS software. In step 3, we analyze various biophysical properties (cluster size distribution, extent of sticker saturation, radial location of stickers inside the condensate etc.) of the system.

1) **Chain connectivity:** To ensure connectivity within a chain, intra-chain beads are connected by harmonic springs ("*bond_style harmonic*" in LAMMPS). Stretching energy of each harmonic bond

$$E_{bond} = K_b \left( R - R_0 \right)^2$$

where $K_b$ is the spring constant and $R_0$ is the equilibrium bond distance. R measures the distance between the bonded beads at any given time. Typical values used in our model, $R_0$ = 10 Å and $K_b$ = 3 $\frac{kcal}{mol*Å^2}$ .

2) **Chain flexibility:** To allow chain flexibility, angle ($\theta$) between three successive beads is modelled with a cosine function ("*angle_style cosine*" in LAMMPS):

$$E_{bending} = \kappa * (1 - cos\theta)$$

where $\kappa$ determines the bending stiffness. Typical value used in our model, $\kappa$ = 2 *kcal. mol*⁻¹.

3) **Specific interaction:** To encode "specific" interactions between complementary sticker types, we introduce reversible bonds (Fig 2B). We use "*fix bond/create/random*", first described here [20], and "*fix bond/break*" functionality



**Fig 2. Quantification of clustering dynamics. (A)** Illustration of the two-component sticker-spacer system. Each component consists of 10 stickers (red and cyan beads) and 40 spacers (yellow beads). We only allow heterotypic interactions, that is, red stickers interact with cyan stickers, but red-red or cyan-cyan are not allowed. **(B)** Stickers engage in specific interactions (Es). A complementary sticker (red and cyan) pair can form a reversible bond within a cutoff radius, Rcut. Once bonded, they cannot engage with another sticker that may be present within Rcut. In other words, each sticker has a valency of 1. **(C)** All beads (except bonded stickers) in the system experience non-specific interactions (Ens), modelled by Lennard-Jones (LJ) potential. One bead can interact with multiple beads, permitted by volume exclusions. A bead diameter ($\sigma$) is set by the minimum distance between two bead centers. **(D)** Snapshots of a multi-chain system undergoing clustering as a function of time. A total of 400 chains (200 chains each type) are placed uniformly inside a cubic box (length = 800Å). Sticker-sticker (Es = 6kT) interactions drive inter-chain crosslinking and weaker spacer-spacer (Ens = 0.3kT) interactions tune the cluster compaction. **(E)** Energy time course of the system. $E_{bond}$ includes all the bonds (permanent and breakable) present in the system. $E_{pair}$ refers to the sum of contact energies coming from the pairwise Lennard-Jones interactions. $E_{angle}$ is angular energy. $E_{potential} = E_{bond} + E_{pair} + E_{angle}$. Energy unit is kcal/mol. **(F)** Time course of radius of gyration ($R_g^{system}$) and sticker saturation. Inset shows the zoomed in version of the first half. The red dashed line indicates the time needed to equilibrate the system's Rg, while the blue dashed line indicates the time needed by the stickers to reach to a steady saturation level. In **(E,F)**, data is averaged over 5 stochastic runs. Solid line is mean, shaded area is standard deviation.

to implement a reversible bond formation scheme. When two stickers approach each other within a cut-off radius ($R_{cut}$), they can form a "bond" with a probability, $p_{on}$. The bond can be broken with a probability of $p_{off}$, if the distance, $R \geq R_{cut}$. These are "specific" interactions because once a sticker pair is bonded, they can't form another bond with

complementary stickers that are still within R_cut. In other words, each sticker has a valency of 1. The inter-sticker bonds are modelled with a shifted harmonic potential ("*bond_style harmonic/shift/cut*" in LAMMPS),

$$E = \frac{E_s}{(R_0 - R_{cut})^2} \left[ (R - R_0)^2 - (R_{cut} - R_0)^2 \right]$$

R is the inter-sticker distance. At the resting bond distance ($R_0$), the energy is $-E_s$. We refer to this well depth parameter as <u>specific energy</u>. When two complementary stickers form a bond, the gain in energy is Es. We also note that, at $R = R_{cut}$, $E = 0$. For $R > R_{cut}$, $E$ is set to be zero. In our model, $R_0 = 1.122 * \sigma$, $\sigma = 10$ Å, $R_{cut} = R_0 + 1.5$Å, $p_{on} = 1$, $p_{off} = 1$.

4) **Non-specific interaction:** Apart from inter-sticker interaction, each pair of beads (stickers + spacers) interacts via a non-bonded isotropic interaction (Fig 2C), modelled by Lennard-Jones (LJ) potential ("*pair_style lj/cut*" in LAMMPS),

$$E_{LJ} = 4 * E_{ns} * \left[ \left(\frac{\sigma}{r}\right)^{12} - \left(\frac{\sigma}{r}\right)^6 \right]$$

where σ represents the bead diameter and r is the separation between the beads. $E_{ns}$ is the depth of LJ energy-well that determines the strength of attractive potential. To distinguish it from specific interaction (described above), we refer to this parameter as <u>non-specific energy</u>. To achieve computational efficiency, the LJ potential is truncated at a cut-off distance ($R_{max}$). In our model, $\sigma = 10$ Å, $R_{max} = 2.5 * \sigma$. We also turn off the LJ interactions for bonded beads with "*special_bonds lj 0 1 1 angle yes*". This also includes stickers which can reversibly switch between bonded and unbonded states.

Our previous studies [7,21] utilized this implementation of Es and Ens, and a precursor form here [6], to probe the biophysics of sticker spacer condensates.

## B. Step 2 - Simulation

**B.1 Langevin dynamics:** We perform Langevin dynamics simulations, in a 3D cuboid volume with periodic boundary conditions, using LAMMPS' "*fix langevin*" thermostat coupled with NVE integrator ("*fix nve*"). We use the "real" unit convention ("*units real*" in LAMMPS) in our simulations. Temperature of the system is usually kept at 310 Kelvin. Mass of each bead can vary depending on the system of interest; a typical value is 100 grams/mole. A typical timestep of 15 to 30 femtoseconds (fs) is used in our simulation. There is a damping ($t_{damp}$) parameter in Langevin thermostat which mimics the viscosity of the medium; typical value, $t_{damp} = 500$ fs. A titration of $t_{damp}$ provides insights on dynamic behaviors (kinetic arrest for condensate size distribution [6], for instance) of the system. The solver attempts to form or break the sticker-sticker cross-linking once in every 20 timesteps.

We generally utilize multiple CPUs to run simulations with MPI-enabled LAMMPS solver. As a rule of thumb, we allocate 500 beads per CPU. A typical system can have $10^4$ to $5*10^4$ beads which may require 20 to 100 CPUs. Since condensate formation can be spatially heterogeneous, we also employ the CPU load balancing strategy ("*comm_style tiled*" & "*fix balance*" in LAMMPS) that adapts a dynamic spatial decomposition of the simulation volume based on the chain density.

**B.2 Metadynamics:** For certain molecular simulation process, standard Langevin dynamics may be a time-consuming strategy. To expedite the process, we employ metadynamics [22,23] ("*fix colvars*" in LAMMPS) which deposits auxiliary gaussian potentials to bias the system to move along the user-defined collective variable (CV). For instance, clustering of chains (process) can be achieved via radius of gyration (CV) axis of the system or merging of two condensates (process) can be achieved via inter-condensate distance (CV) axis [7].

## C. Step 3 - Data analysis

Once the LAMMPS simulations are over, we can analyze the results guided by questions of interest. Since Langevin dynamics has a stochastic component, we typically run 5 trials and sample the output to obtain statistically robust behavior. A multitude of analysis sheds light on the kinetic and thermodynamic state of the system. Time course of the potential energy informs us about whether the system converges to a steady state or not. How the bonded sticker fraction evolves with time is another important variable to analyze. During the course of simulation, we also save the system configuration ("system_timestamp.restart") at some regular interval. These restart files contain information about the location and connectivity of chains which can be converted into a network. Extending the capabilities of MolClustPy [24], we then employ graph-based analysis to identify subnetworks at a given timepoint and to examine how the sub-network

distribution evolves over time. Since a sub-network is a cluster of connected chains, this analysis provides detailed statistics of cluster size distribution and extent of sticker saturation within those clusters. This method can also be used to extract spatial topology of the cluster; one example is to compute the radial density profile of stickers and spacers within a cluster and how it changes as a function of sticker pattern in a chain. In the next section, with example simulations, we demonstrate the key strengths of our pipeline.

# Results and discussion

## 1. Dynamics of condensate formation

We consider a two-component heterotypic system (Fig 2A) where complementary stickers from different chains form crosslinks and drive clustering. Each chain contains a total of 50 beads where 10 stickers are distributed uniformly along the sequence. Red-stickers from chain type A can interact with cyan-stickers from chain type B. Specific energy (Fig 2B), Es = 6kT and non-specific energy (Fig 2C), Ens = 0.3kT.

Fig 2D displays temporal snapshots of the system as the chains undergo clustering, while Fig 2E reports the corresponding energy trend that gradually drops and reaches a plateau. The potential energy has three components – bonded ($E_{bond}$), non-bonded ($E_{pair}$) and angular ($E_{angle}$) energy. As more stickers form reversible bonds, bonded energy (blue line) drops since each inter-sticker bond lowers the energy by Es (6kT in this case). The $E_{pair}$ trace closely follows $E_{bond}$, but with a slight delay (magenta line), which indicates that the spacer mediated interactions kick in once the specific bonds hold the chains together. This secondary interaction controls the density of the cluster. The slight downward trend in angular energy (yellow line) shows relaxations of individual chains once they are fully integrated within the cluster. At an earlier stage, there are more smaller clusters with higher surface area which put angular strains to more chains – an effect akin to surface tension. When those segments merge to become one large spherical cluster, due to reduction in surface-to-volume ratio, more chains adopt relaxed conformations with an overall drop in angular energy.

Fig 2F highlights the interplay of two processes – sticker saturation and clustering of chains. Sticker saturation is defined as the fraction of stickers that are in bonded state. $R_g^{system}$ is the radius of gyration computed for all the beads in the system. Around 40 million steps, sticker saturation reaches the maximal level (~ 0.72) while the system takes about 170 million steps to reach the highest compaction or lowest $R_g^{system}$. This is consistent with the slight delay in time trace between $E_{bond}$ and $E_{pair}$ (Fig 2E). At Es = 6kT and Ens = 0.3kT, sticker-sticker bonds act as primary drivers of clustering; spacer interactions operate on a secondary level and make the clusters more compact. We note here that spacers alone can't trigger clustering, and the chains will remain in a dispersed state in the absence of stickers. This provides a biological control mechanism where phase separation can be reversibly induced by changing the sticker state (on and off). More importantly, the kinetic interplay of sticker saturation and clustering can lead to non-trivial outcomes as a function of energy parameters (Es, Ens) and simulation volume. For stronger Es and larger simulation box, chains will collapse into smaller clusters with high degree of saturated intra-chain stickers; this yields a metastable system [6,7] with many long-living clusters instead of one large cluster.

## 2. Specific interaction scheme conforms to the detailed balance

Next, we set out to test whether our specific interaction scheme (between stickers) conforms to the principle of microscopic reversibility a.k.a. detailed balance. When two stickers form a bond, the system gains Es amount of energy; breaking that bond requires overcoming the energy well (Figs 3A and 2B). If we consider a two-state (bound: B and unbound: U) system, then requirements of detailed balance read:

$$\pi_U P_{U\to B} = \pi_B P_{B\to U} \tag{1}$$

$$K_{eq} = \frac{\pi_U}{\pi_B} = \frac{P_{B\to U}}{P_{U\to B}} \propto e^{-\frac{E_s}{kT}} \tag{2}$$

[$\pi_U$, $\pi_B$] indicates stationary probabilities of finding the system in unbound and bound states, respectively. $P_{U\to B}$ corresponds to the unbound-to-bound transition rate, while $P_{B\to U}$ reflects the reverse rate. Equation 1 denotes the equality of fluxes. Equation 2 posits that equilibrium configuration of such two-state system should obey the Boltzmann distribution.

To test out the validity of detailed balance criteria, we set up a two-particle simulation (Fig 3B) in a pseudo 1D box where two stickers can diffuse and form a bond reversibly. Since there are only two stickers, the system can only switch between bond=0 and bond=1 (Fig 3C). To quantify switching frequency, we capture the system at two successive timeframes which displays one of the four configurations: BB, BU, UB and UU. Collecting these statistics from a long

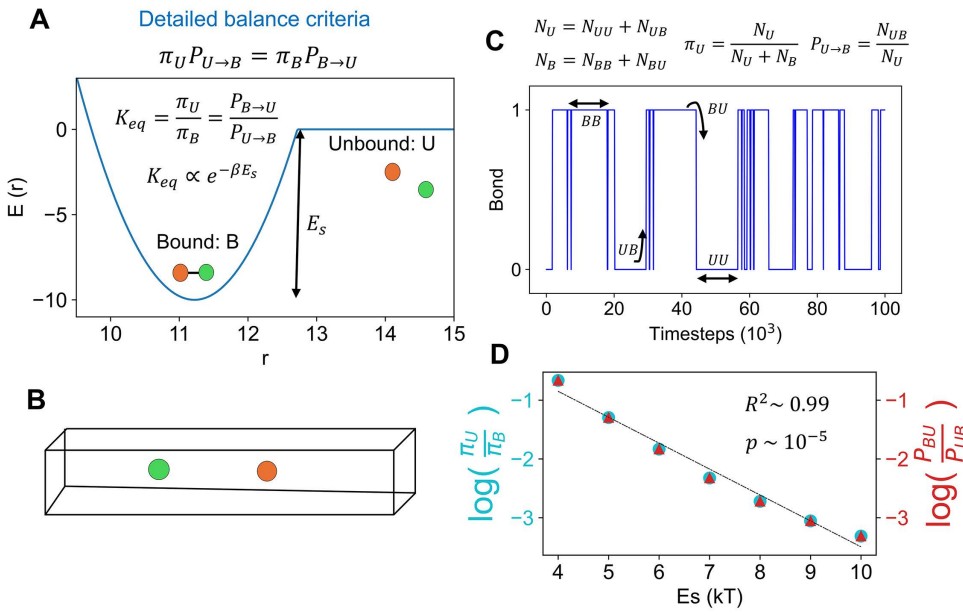

**Fig 3. Test of detailed balance for specific interactions. (A)** Thermodynamic criteria of detailed balance for a two-sticker (orange and green) specific interaction scheme. $\pi_U$ and $\pi_B$ are stationary probabilities of finding the system in unbound (U) and bound (B) state, respectively. $P_{B\to U}$ and $P_{U\to B}$ are transition rates from bound to unbound, and vice versa. $K_{eq}$ is equilibrium constant that varies exponentially with the depth of the energy well (Es). **(B)** Simulation setup consisting of two stickers. The expediate the diffusion driven search process, a pseudo-one-dimensional box (with periodic boundaries) is considered where both stickers can switch between bonded and unbonded states. **(C)** A representative time course of bonding dynamics (1 is bonded and 0 is unbonded). At two successive frames (t and t'), the system (s) can display one of the four possibilities; $BB : s[t] = 1$ and $s[t'] = 1$, $BU : s[t] = 1$ and $s[t'] = 0$, $UB : s[t] = 0$ and $s[t'] = 1$, $UU : s[t] = 0$ and $s[t'] = 0$. **(D)** Scaling behavior of equilibrium quantities with specific interaction strength, Es. Two sides of vertical axis show two different quantities (color coded). The black dashed line is the linear fit. A linear regression with R²=0.99 and a p-value of 2×10⁻⁵ indicates an excellent linear fit.

trajectory (10 million frames), we can extract stationary and dynamic information of the system by the following method: $\pi_U = \frac{N_U}{N_U + N_B}$, $\pi_B = \frac{N_B}{N_U + N_B}$, $P_{U \to B} = \frac{N_{UB}}{N_U}$, $P_{B \to U} = \frac{N_{BU}}{N_B}$, where $N_U = N_{UU} + N_{UB}$, $N_B = N_{BB} + N_{BU}$. $N_{ij}$ refers to the number of frames where the system can be found in state i and j at two successive timeframes.

Fig 3D summarizes the results from the two-particle simulations. There are two major findings:

a. for a given Es, the cyan circle perfectly coincides with the red triangle, that is, $\frac{\pi_U}{\pi_B} = \frac{P_{B \to U}}{P_{U \to B}}$ => $\pi_U P_{U \to B} = \pi_B P_{B \to U}$. This proves the validity of first criteria shown by Equation 1.

b. the logarithm of thermodynamic constant drops linearly with Es, that is, $\log(K_{eq}) \propto -E_s$ => $K_{eq} \propto e^{-E_s}$. This is the second validation as outlined in Equation 2.

Hence, the two-particle simulation provides evidence in favor of the detailed balance. The reversible bond-formation protocol between stickers seems to be consistent with the basic tenets of statistical physics.

### 3. Phase transition boundary and bond lifetime

One major application of CASPULE is to probe the phase transition behavior of biopolymers. To demonstrate an example (Fig 4A), we consider a preformed cluster (as formed in Fig 2D, last snapshot) and titrate the specific interaction energy (Es) between stickers. At a low Es, the cluster dissolves (Fig 4A). Above a threshold level of Es, the cluster is stable. If we systematically compute the clustering state of the system as a function of Es, we observe a switch-like behavior (Fig 4B).

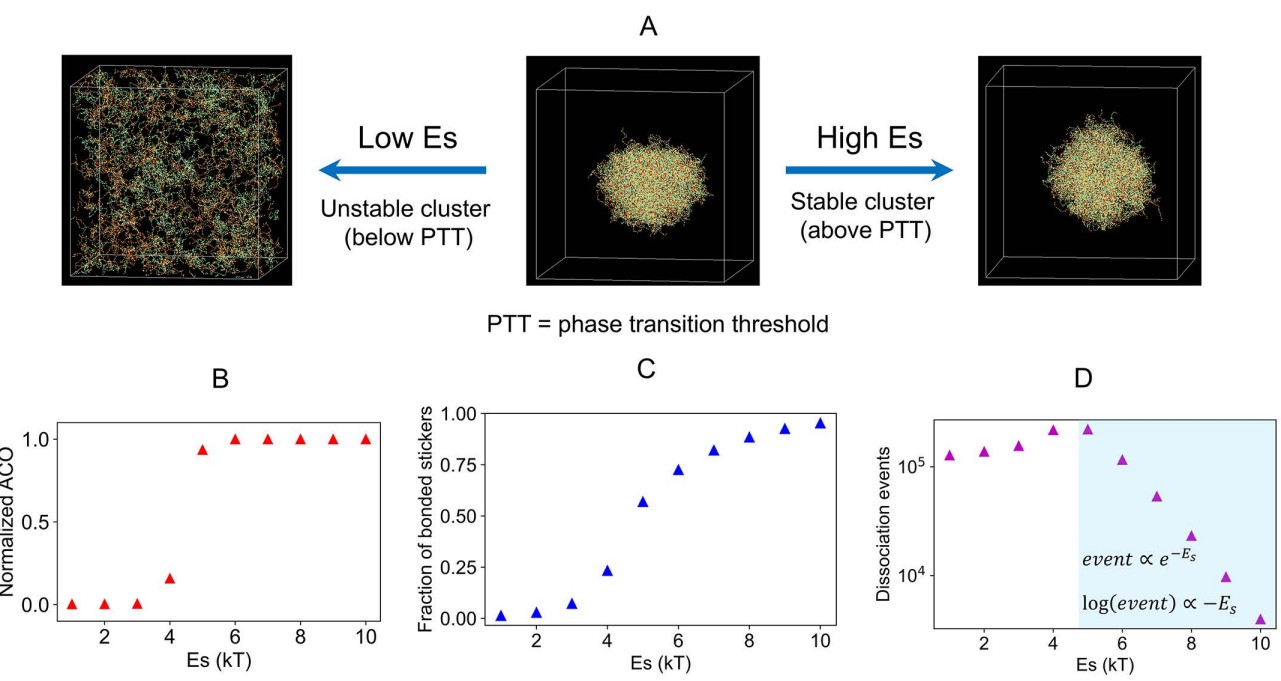

**Fig 4. Quantification of the phase transition boundary. (A)** Illustration of phase transition by titrating specific energy (Es). The cluster dissolves at Es lower than a critical value. Above the critical level of Es, cluster is stable. **(B)** Average cluster occupancy (ACO) normalized by the total number of chains in the system. See main text for definition. **(C)** Fraction of bonded sticker $\left(\frac{Bonded\ stickers}{Total\ stickers}\right)$ or sticker saturation. **(D)** Inter-sticker dissociation events. The shaded area indicates a region after the phase transition.

Here, ACO measures the average clustering state of the system, as introduced here [24]. The fraction of total chains in the cluster of size $S_i$,

$$f_i = \frac{n_i S_i}{N_{total}}$$

where $n_i$ is the number of clusters of size $S_i$ and $N_{total}$ is the total chain count. Then,

$$ACO = \sum_i f_i S_i = \frac{1}{N_{total}} \sum_i n_i S_i^2$$

ACO is then normalized by the total chain count. The upper limit of this normalized parameter is now 1, which includes all chains in one large cluster. When the system is fully dispersed (all chains are in monomeric state), $ACO \sim \frac{1}{N_{total}} \ll 1$. We note that ACO can provide useful information once the system contains a mix of large clusters. This quantification of cluster distribution is important as biological systems often display a size distributed state that may be functionally critical [21]. In our current system, we are dealing with a transition between the dispersed state and a single-cluster state. The sharp change in ACO (Fig 4B) indicates a phase transition where the energy threshold lies around Es = 4kT. When we look at the bonded sticker fraction (Fig 4C), it also shows sigmoidal behavior but with more gradual change. Nevertheless, a critical fraction of stickers needs to be bonded to trigger the phase transition which is cooperative in nature. It is important to note that the numerical accuracy and execution time of our simulations depends on the timestep (S1 Fig). Shorter timesteps improve the accuracy of results while delaying the execution time. The viscosity (damping time) parameter in Langevin dynamics does not affect the potential energy (S2 Fig) of the system. However, it controls the kinetics by tuning the diffusion coefficients (S3 Fig) in an expected manner consistent with the Einstein-Stokes relation. It is worth highlighting that the diffusivity-mediated kinetic effect plays an important role in controlling the size of metastable condensates [6] where chains get arrested in smaller droplets in a viscosity dependent manner. In the current work, we overcome this barrier by starting the simulations in a smaller volume (Fig 2D). However, with larger box size and identical Es, the sticker saturation kinetics remains the same, but the timescale related to diffusion increases – this leads to multiple smaller droplets (kinetic arrest) in place of a single large droplet (equilibrium-like). Such effect is not expected to emerge in a slab-like geometry (which is a commonly used technique to access the two-phase equilibrium) since diffusion in three-dimensional space is qualitatively different than the one observed in "pseudo-one-dimensional" geometry.

We observe another interesting trend when we compute the inter-sticker dissociation events for a given time interval ($10^6$ steps in this case). Fig 4D shows a non-monotonic trend of events as we titrate up the Es. Before the phase transition, chains majorly remain as monomers and small oligomers. In this regime, lifetime of individual bond is short, hence we get more dissociation events. After phase transition, the dissociation events gradually go down with Es. Importantly, on a logarithm scale, the decay trend is linear. This suggests that sticker dissociation kinetics follow an Arrhenius-like form, *event* $\propto e^{-E_s}$. Since inter-sticker bond formation is diffusion-limited in nature (*pOn* = 1 *when* $R \leq R_{cut}$) and bond dissociation requires overcoming the harmonic energy well (depth is Es), this Arrhenius-like expression extends the validity of the detailed balance criteria (Fig 3) for this many-particle system.

## 4. Effect of system size on simulation time

Probing condensate biophysics often requires variation of the system size in terms of the simulation volume or molecular counts. Fig 5A shows a scenario where we systematically create clusters with different sizes ($N_{chain}$) and simulate the system, in parallel, with different numbers of CPUs. The MPI enabled LAMMPS solver can efficiently simulate a system across multiple CPUs. In this computational experiment, we first place N/2 chains of each type ($N_A + N_B = N_{chain} = 40, 80, 120, 160, 200, 400, 600, 800$) in a smaller box and let them cluster. The fully formed clusters

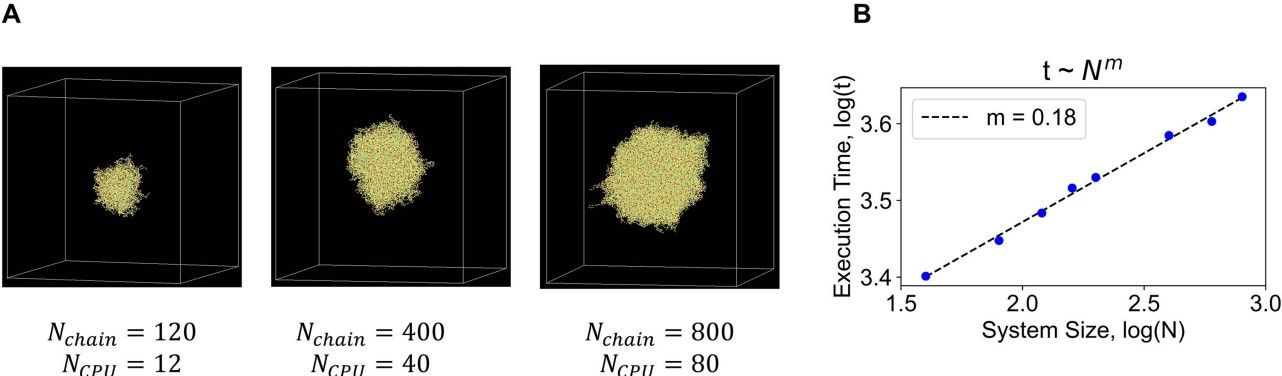

**Fig 5. Scaling of execution time with system size. (A)** Three representative clusters with different chain counts ($N_{chain}$) that are simulated with higher number of CPUs ($N_{CPU}$) in parallel. **(B)** Execution time (reported in seconds, log10 scale) as a function of the system size.

are then placed in a larger box (L = 120nm) and simulated for 10 million steps with a proportionately higher number of CPUs (Fig 4A). For example, cluster of 120 chains with 12 CPUs and cluster of 800 chains with 80 CPUs. We also should highlight that the load_balance feature ("*comm_style tiled*" & "*fix balance*") of LAMMPS is enabled here which allows dynamic spatial decomposition of the simulation volume and allocate CPUs based on spatial density. Family of CPUs used in these simulations belong to "Intel Xeon Platinum 8480CL" (sapphire partition in Harvard FAS RC HPC, as of October 2025).

Fig 5B shows the execution time trend as we increase the system size. The data is plotted on a log-log scale. Let's say, the execution time (t) scales with system size (N) by a power-law, $t \sim N^m$. For a linear regime, m = 1 where time to simulate a system grows proportionately with the size of the system. On the other end, due to parallelization, if execution time is insensitive to system size, then m = 0. To our satisfaction, we find that m = 0.18 for our simulations. This smaller exponent verifies the scalability of our approach provided computational resource ($N_{CPU}$) is not a constraint.

## Summary

Embedded in the powerful and popular MD engine (LAMMPS), CASPULE is a versatile and reproducible protocol to simulate and analyze biological condensates. In this pipeline, single valent sticker interactions drive polymer cross-linking and weaker non-specific interactions aid in cluster compaction. The reversible bond formation between stickers is consistent with the detailed balance criteria, and the lifetime of inter-sticker bonds decays exponentially with the interaction energy, Es. Due to parallel execution across multiple CPUs, our approach is efficient and scalable that aims to bridge the gap between traditional MD (molecular dynamics) and MC (Monte Carlo) based solvers. Our goal here is to give researchers an MD framework rooted in a mainstream engine that treats stickers properly with limited valency. We believe that our framework makes it practical to explore design rules for condensate formation, stability, composition, and function in scenarios where directionality and valency of interactions is a key governing factor for phase separation.

## Supporting information

**S1 Fig. Effect of timestep.** Trend of (A) potential energy (B) Kinetic energy (C) Sticker bonding statistics and (D) execution time. Energy parameter, Es = 6kT, Ens = 0.3kT. Damping parameter = 200 fs. Total elapsed or simulation time = $50 * 10^6$. (TIF)

**S2 Fig. Effect of damping time.** Trend of the potential energy at three different values of damping time. Energy parameter, Es = 6kT, Ens = 0.3kT. Damping parameter = 200 fs. Total elapsed or simulation time = $50 * 10^6$.
(TIF)

**S3 Fig. Effect of damping time on cluster diffusion.** (A) Mean squared displacement vs. elapsed time plot for six different damping time (viscosity). The circles indicate actual displacements, and solid lines show the linear fit. Diffusion coefficient, D, is extracted from the linear fit: MSD = 6Dt (B) D vs. damping time in log-log scale. The red points are extracted Ds from MSD trajectories. The black dashed line is the linear fit which yields a slope ~ 1. In these simulations, timestep = 10 fs. Energy parameters, Es = 6kT, Ens = 0.3kT.
(TIF)

**S1 Appendix. Supplementary texts on the effect of timestep and viscosity on simulation accuracy.**
(PDF)

## Acknowledgments

We are thankful to Harvard FAS-RC High Performing Computing facility.

## Author contributions

**Conceptualization:** Aniruddha Chattaraj, Eugene I. Shakhnovich.

**Data curation:** David S. Kanovich.

**Formal analysis:** Aniruddha Chattaraj.

**Funding acquisition:** Eugene I. Shakhnovich.

**Investigation:** Aniruddha Chattaraj, Eugene I. Shakhnovich.

**Methodology:** Aniruddha Chattaraj, David S. Kanovich, Srivastav Ranganathan.

**Project administration:** Eugene I. Shakhnovich.

**Resources:** Srivastav Ranganathan.

**Software:** Aniruddha Chattaraj, David S. Kanovich.

**Supervision:** Eugene I. Shakhnovich.

**Validation:** Aniruddha Chattaraj, Eugene I. Shakhnovich.

**Visualization:** Aniruddha Chattaraj, David S. Kanovich.

**Writing – original draft:** Aniruddha Chattaraj, David S. Kanovich, Srivastav Ranganathan.

**Writing – review & editing:** Aniruddha Chattaraj, Eugene I. Shakhnovich.

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
