## [Decision Letter · Decision Letter 0]

5 Jan 2026

PCOMPBIOL-D-25-02356

CASPULE: A computational tool to study sticker spacer polymer condensates

PLOS Computational Biology

Dear Dr. Shakhnovich,

Thank you for submitting your manuscript to PLOS Computational Biology. After careful consideration, we feel that it has merit but does not fully meet PLOS Computational Biology's publication criteria as it currently stands. Therefore, we invite you to submit a revised version of the manuscript that addresses the points raised during the review process.

We look forward to receiving your revised manuscript.

Kind regards,

Tamar Schlick

Academic Editor

PLOS Computational Biology

Nir Ben-Tal

Section Editor

PLOS Computational Biology

**Additional Editor Comments:**

Thank you for your submission.

While the software appears useful, two reviewers

make many comments regarding the methodology

and writing that need to be addressed thoroughly before

further consideration. Concerns regarding the rigour of the

methodology and claims made on its properties

need to be fully justified. The software details must also

be written clearly so users can follow the information easily.

**Journal Requirements:**

3) Some material included in your submission may be copyrighted. According to PLOSu2019s copyright policy, authors who use figures or other material (e.g., graphics, clipart, maps) from another author or copyright holder must demonstrate or obtain permission to publish this material under the Creative Commons Attribution 4.0 International (CC BY 4.0) License used by PLOS journals. Please closely review the details of PLOSu2019s copyright requirements here: PLOS Licenses and Copyright. If you need to request permissions from a copyright holder, you may use PLOS's Copyright Content Permission form.

Potential Copyright Issues:

i) Figure 1. Please confirm whether you drew the images / clip-art within the figure panels by hand. If you did not draw the images, please provide (a) a link to the source of the images or icons and their license / terms of use; or (b) written permission from the copyright holder to publish the images or icons under our CC BY 4.0 license. Alternatively, you may replace the images with open source alternatives. See these open source resources you may use to replace images / clip-art:

4) Please amend your detailed Financial Disclosure statement. This is published with the article. It must therefore be completed in full sentences and contain the exact wording you wish to be published.

5) Kindly revise your competing statement in the online submission form to align with the journal's style guidelines: 'The authors declare that there are no competing interests.'

**Reviewers' comments:**

Reviewer's Responses to Questions

**Comments to the Authors:**

Reviewer #1: The authors describe a simulation pipeline to perform off-lattice sticker-spacer simulations of binary heteropolymer mixtures, in which reactive beads on one polymer species can form "specific" one-to-one bonds with reactive beads on the other polymer species. Although simulations of this system by this group and others (e.g., Pappu, Wingreen) have been reported previously, including using this off-lattice approach, the present manuscript describes the steps required to set up and run such simulations clearly. It therefore reads as a user-friendly extended methods section for previous papers on this system from this group. Overall, the manuscript is well organized and easy to read.

I have two scientific concerns regarding the methodology:

1. It is not clear that the "specific interaction" scheme, which is based on "fix bond/break", satisfies detailed balance. First, the authors should provide a proof, both because this property is essential for any simulation method and also because the pedagogical nature of this methods-oriented manuscript requires that all algorithmic details are completely explained. Second, the authors should show that the results (e.g., phase boundaries) are insensitive to the time step and damping factor in the Langevin simulation. Third, the authors should show that these results are invariant to changes in the integration scheme, e.g., to Nose-Hoover or even to a Metropolis Monte Carlo algorithm. Finally, the authors should show that no issues arise from combining Monte Carlo bond breaking/formation moves with Langevin dynamics, since it is well known that subtle problems arise from non-area-preserving integration schemes (see, e.g., Bussi, Giovanni, and Michele Parrinello. "Accurate sampling using Langevin dynamics." Physical Review E—Statistical, Nonlinear, and Soft Matter Physics 75.5 (2007): 056707.).

2. It is standard practice to use slab-geometry direct-coexistence simulations to compute phase boundaries, since the curved interfaces of finite-size droplets can modify the coexistence conditions due to a non-zero Laplace pressure. Since the manuscript discusses the calculation of phase boundaries, results using the slab geometry should be presented.

Reviewer #2: This manuscript presents CASPULE, a computational pipeline for simulating sticker–spacer polymers with strict single-valent bonding using Langevin dynamics in LAMMPS. Overall, I find this to be a reasonable and potentially useful contribution. The work clearly demonstrates how limited-valency interactions can be incorporated into particle-based simulations, and such interactions are increasingly recognized as important for modeling biomolecular condensates. Compared to existing lattice-based or Monte Carlo approaches, the ability to access dynamical information is a notable strength and could be valuable for studying kinetic arrest, coarsening, and aging phenomena in condensates.

That said, in its current form the project feels closer to a detailed simulation protocol than to a polished, user-friendly software package. While the documentation is thorough, the setup requires users to navigate multiple external tools (e.g., Moltemplate, PACKMOL, custom LAMMPS scripts) and manually coordinate several steps. This creates a fairly high barrier to entry. Streamlining the workflow, ideally by wrapping the setup and execution steps into a unified Python interface, would substantially increase the accessibility and impact of CASPULE for the broader community.

In addition, the manuscript emphasizes access to dynamical behavior, but the reversible bonding scheme relies on fix bond/create and fix bond/break in LAMMPS. To my knowledge, these fixes do not rigorously satisfy detailed balance, and therefore do not guarantee that the resulting dynamics (or even steady-state distributions) are thermodynamically consistent. This does not invalidate the approach, but it does raise important questions about how faithfully the reported dynamics correspond to an underlying physical model. A more explicit discussion of this issue, along with its possible implications for interpreting kinetic rates, bond lifetimes, and steady-state properties, would strengthen the manuscript and help readers better understand the scope and limitations of the method.

Overall, I believe CASPULE has clear potential, but addressing the points above would significantly improve both its usability and the clarity of its physical interpretation.

Reviewer #3: The author provide a tool that should be useful for simulations of phase separation. Although I have not testet it extensively it seems to work well, documentation is nice and the paper is well written

**Have the authors made all data and (if applicable) computational code underlying the findings in their manuscript fully available?**

Reviewer #1: Yes

Reviewer #2: Yes

Reviewer #3: Yes

PLOS authors have the option to publish the peer review history of their article (what does this mean?). If published, this will include your full peer review and any attached files.

Reviewer #1: No

Reviewer #2: No

Reviewer #3: **Yes:** Arne Elofsson

**Figure resubmission:**

After uploading your figures to PLOS’s NAAS tool - https://ngplosjournals.pagemajik.ai/artanalysis, NAAS will process the files provided and display the results in the "Uploaded Files" section of the page as the processing is complete. If the uploaded figures meet our requirements (or NAAS is able to fix the files to meet our requirements), the figure will be marked as "fixed" above. If NAAS is unable to fix the files, a red "failed" label will appear above. When NAAS has confirmed that the figure files meet our requirements, please download the file via the download option, and include these NAAS processed figure files when submitting your revised manuscript. **Reproducibility:**

---

## [Decision Letter · Decision Letter 1]

29 Apr 2026

Dear Prof. Shakhnovich,

We are pleased to inform you that your manuscript 'CASPULE: A computational tool to study sticker spacer polymer condensates' has been provisionally accepted for publication in PLOS Computational Biology.

if possible, please attend to remaining comment of Reviewer 1 before submitting your final files.

Best regards,

Tamar Schlick

Academic Editor

PLOS Computational Biology

Nir Ben-Tal

Section Editor

PLOS Computational Biology

If possible, attend to remaining comment when you submit your final files.

Reviewer's Responses to Questions

**Comments to the Authors:**

Reviewer #1: The authors have addressed my comments from the previous round of review. I recommend publication given that this tool is likely to be of use to the community. I still would encourage the authors to push a bit further with the detailed balance verification: Checking the results for a single dimer is an appropriate test, and the rationale is clearly explained in the revised text, but the validation could be more quantitative. The equilibrium bound fraction can be computed exactly from a relatively simple integral involving the potential, as opposed to extracting it from simulation, thus establishing the precise expected dependence of log(pi_U/pi-B) on Es. It would be more convincing to demonstrate that the simulation reproduces this known result for all values of Es within the sampling error. This would also eliminate any uncertainty regarding the apparent systematic curvature of the fit shown in the new figure 3D.

Reviewer #2: The authors have successfully addressed my previous comments and I support the publication of the manuscript.

**Have the authors made all data and (if applicable) computational code underlying the findings in their manuscript fully available?**

Reviewer #1: Yes

Reviewer #2: None

PLOS authors have the option to publish the peer review history of their article (what does this mean?). If published, this will include your full peer review and any attached files.

Reviewer #1: No

Reviewer #2: No